# Assessing the impact of the COVID-19 pandemic on uptake and experiences of gestational diabetes mellitus screening in Ontario: A parallel convergent mixed-methods study

Dima Hadid[1☯], Rebecca H. Correia[1,2☯], Sarah D. McDonald[3], Elizabeth K. Darling[4], David Kirkwood[5], Aaron Jones[2], Andrea Carruthers[1], Cassandra Kuyvenhoven[1], Michelle Howard[1], Devon Greyson[6], Sujane Kandasamy[7,8], Meredith Vanstone[1]*

1 Department of Family Medicine, McMaster University, Hamilton, Canada, 2 Department of Health Research Methods, Evidence and Impact, McMaster University, Hamilton, Canada, 3 Department of Obstetrics and Gynecology, Division of Maternal-Fetal Medicine, McMaster University, Hamilton, Canada, 4 Department of Obstetrics and Gynecology, McMaster University, Hamilton, Canada, 5 ICES, McMaster University, Hamilton, Canada, 6 School of Population and Public Health, University of British Columbia, Vancouver, British Columbia, Canada, 7 Department of Child and Youth Studies, Brock University, St. Catherine's, Canada, 8 Department of Medicine, McMaster University, Hamilton, Canada

☯ These authors contributed equally to this work.
* vanstomg@mcmaster.ca

**Data Availability Statement:** Qualitative data are not available for sharing; when participants

## Abstract

### Objective

Gestational diabetes mellitus (GDM) is a common medical complication of pregnancy that leads to adverse outcomes for both infants and pregnant people. Early detection and treatment can mitigate these negative outcomes. The COVID-19 pandemic strained healthcare and laboratory services, including GDM screening programs. Adapted GDM screening guidelines were introduced in many jurisdictions. This study examined changes in uptake, modality, and experiences of GDM screening in Ontario, Canada during the COVID-19 pandemic.

### Methods

This convergent mixed-method study involved a population-based retrospective cohort analysis of Ontario-based health administrative data to describe and compare gestational diabetes screening rates among 85,228 individuals with live, in-hospital births between January 1-March 31 before (2019) and during the COVID-19 pandemic (2021 and 2022). Descriptive analyses were conducted for GDM screening pathways aligning with usual and pandemic-adapted screening guidance. Qualitative descriptive interviews were conducted about experiences and decision-making of GDM screening with 43 Ontario residents who gave birth between May 2020 and December 2021. Data were integrated during the design and interpretation phases.

consented to participate in the study, they did not consent to public access to transcripts or data beyond what is reported within this report. The dataset for the quantitative portion of this study is held securely in coded form at ICES. While legal data sharing agreements between these data stewards and data providers prohibit ICES from making the dataset publicly available, access may be granted to those who meet pre-specified criteria for confidential access. Readers are welcome to contact the research team for further information. Data queries to ICES can be directed to Data or Analytic Services at das@ices.on.ca.

**Funding:** This study was funded by the Canadian Institutes of Health Research grant number 179921. MV is supported by a Canada Research Chair in Ethical Complexity in Primary Care. SM is supported by a Canada Research Chair, Tier II. DG is supported by a CIHR/PHAC Applied Public Health Chair and Michael Smith Health Research BC Scholar Award. This study was supported by ICES, which is funded by an annual grant from the Ontario Ministry of Health (MOH) and the Ministry of Long-Term Care (MLTC). The funders had no role in study design, data collection and analysis, decision to publish, or preparation of the manuscript.

**Competing interests:** The authors have declared that no competing interests exist.

## Results

There were small but significant increases in GDM screening during the pandemic; likelihood of screening completion using any modality increased in 2021 and 2022 compared to 2019. Testing modality shifted; the alternate screening strategies introduced during COVID-19 were adopted by clinicians. Interview participants perceived GDM screening to be important and obligatory but accompanied by a degree of stress about potential COVID-19 exposure.

## Conclusion

Despite health system challenges experienced in Ontario during the COVID-19 pandemic, GDM screening rates increased in the study population, demonstrating the success of adapted GDM screening guidelines. Decisions about screening modalities were driven by clinician expertise, and interview participants were satisfied to provide informed consent to these recommendations.

## Introduction

Gestational diabetes mellitus (GDM) is a common medical complication of pregnancy that can lead to substantial short and long-term adverse health effects to both the pregnant person and fetus [1]. Unmanaged GDM increases the risk of gestational hypertension, pre-eclampsia, birth-related injuries, stillbirth, post-surgical infections, fetal macrosomia, neonatal hypoglycemia, and type 2 diabetes mellitus (T2DM) [2–5]. The epidemiology of GDM shows significant variation across the world due to variability in risk factors, diagnostic criteria, and measurement. Globally, reported rates of GDM range from as low as 1% to more than 30% [3]. In Canada, where this study was conducted, GDM is on the rise, with current research estimating the incidence of GDM at approximately 7 to 14.9% [4–6].

In 2018, Canadian clinical practice guidelines recommended that every pregnant person be screened for GDM between 24 and 28 weeks of pregnancy (Fig 1); the suggested testing modality was a 50-gram oral glucose challenge test (OGCT) followed by a 75-gram oral glucose tolerance test (OGTT) in individuals who exceeded predetermined blood glucose thresholds [7]. When the COVID-19 pandemic began in March 2020, many countries adapted their GDM screening guidelines in response to the unprecedented challenges to healthcare systems (e.g., fewer in-person appointments, medical laboratories at capacity, resource redeployment) [8–10]. Adapted guidelines released jointly by the Diabetes Canada and the Society of Obstetricians and Gynecologists of Canada recommended that physicians choose a screening modality based on the risk level of the individual patient alongside consideration of the local health system's COVID-19 burden [11,12]. During times of severe health system disruptions, the adapted guidelines recommend that patients without increased risk be screened via a blood test (HbA1c) together with a non-fasting random plasma glucose test [11]. Those with a result considered low risk from this initial screen required no further testing, while those with a high-risk result (defined as an HbA1c >5.7% or random plasma glucose >11.1%) were advised to complete an OGCT and then an OGTT [11]. In contrast, the guidelines recommended that persons with elevated risk for GDM, amidst moderate health system disruptions, be screened with an OGCT followed by an OGTT if warranted [12]. Fig 1 illustrates the traditional and adapted screening pathways.

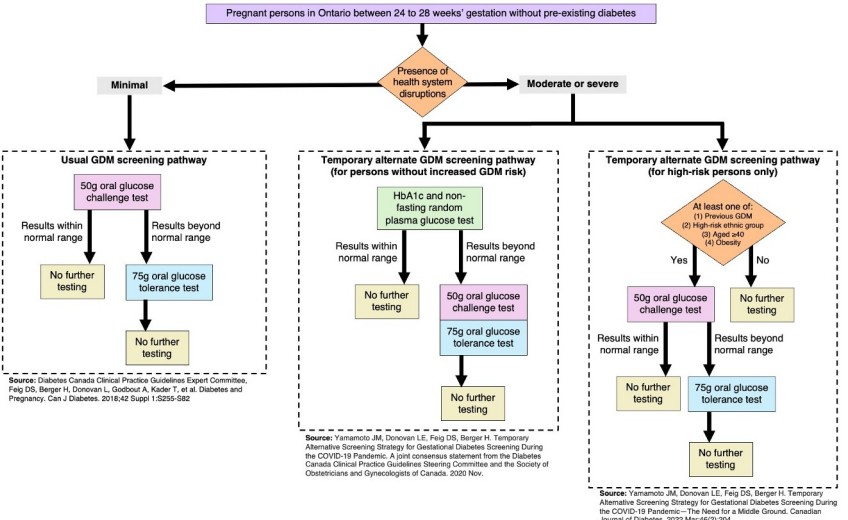

**Fig 1. Usual and alternate gestational diabetes screening pathways.** GDM = gestational diabetes mellitus. Legend: Each pathway ending with "no further testing Indicates that there are no further recommended screening modalities. This figure does not specify the clinical criteria warranting GDM diagnoses, which is determined at the point of "no further testing".

Both the OGCT and OGTT require blood samples to be collected before and after consuming a glucose drink. These tests require waiting for follow-up for 1 and 2 hours, respectively. In contrast, the HbA1c and non-fasting random plasma glucose tests require only minutes to complete. This shorter duration of time was intended to increase screening accessibility while reducing potential in-person COVID-19 exposure [11]. Although the higher specificity and lower sensitivity of the HbA1c test may be less effective in detecting GDM, it was deemed adequate to detect individuals at the highest risk [11].

After these guidelines were introduced in April 2020, the burden of COVID-19 on the healthcare system fluctuated. The uptake and modality of GDM screening in Canada over this period is unknown, although there is evidence that GDM screening participation slightly decreased [13] and rates of GDM diagnosis increased in other countries [14,15]. Further, it is unknown how pregnant people in Canada experienced GDM screening during the pandemic, specifically their experience of an adapted screening pathway. This convergent mixed-methods study aims to understand how pregnant people experienced GDM screening during the pandemic, with an emphasis on adapted testing modalities.

## Methods

### Study design

We conducted a convergent parallel mixed methods study to compare GDM screening rates before and during the COVID-19 pandemic. We integrated qualitative data about the experience of GDM screening participation in 2020 and 2021 with quantitative data during the design and interpretation stages (Fig 2). The protocol for this study has been published [16], and it was prospectively registered with clinicaltrials.gov as NCT05663762. The current manuscript reports only on data from Ontario, Canada due to health administrative data availability. We adopted a descriptive qualitative approach to ensure explicit representation of participants' viewpoints [17]. Our reporting adheres to the guidelines, Strengthening the Reporting of

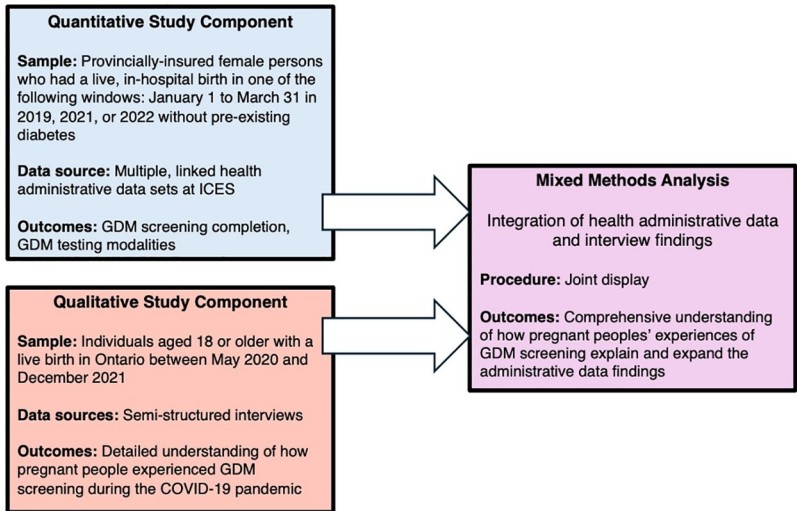

**Fig 2. Overview of convergent parallel mixed methods design.** GDM = gestational diabetes mellitus.

Observational Studies in Epidemiology (STROBE) Statement and Standards for Reporting Qualitative Research (SRQR) (S1 File).

## Setting

Ontario, Canada has a publicly funded healthcare system that provides residents with access to GDM testing [18]. Ontario experienced a significant COVID-19 disease burden, which had a sustained impact on its healthcare system [19].

## Quantitative strand

We conducted a population-based retrospective cohort study to answer the following research question: *Were rates of GDM screening and testing modalities different between persons who gave birth in 2019, 2021, and 2022?* We hypothesized that those who gave birth in 2021 would have lower rates of gestational diabetes screening for any modality than those who gave birth in 2019 or 2022, although adapted screening modalities would be higher for those who gave birth in 2021 and 2022.

**Data sources and study population.** Population-based administrative health data were acquired from ICES, an independent non-profit research institute whose legal status under Ontario's health information privacy law allows it to collect and analyze health care and demographic data without consent for health system evaluation and improvement. The study population included provincially insured individuals who had a live, in-hospital birth during the following periods: January 1 to March 31, 2019, 2021, or 2022. Childbirth was assigned as our index event, with a lookback window spanning 40 weeks prior representing gestation (S2 File). Individuals with pre-existing diabetes before the conception date, those diagnosed with diabetes during gestation (prior to the index event), and persons with a gestational age <20 weeks were excluded from this study. We identified our cohort using the MOMBABY dataset, which links inpatient hospital records of newborns and pregnant persons. Descriptions of all datasets are provided in S3 File; our cohort creation details have been previously published [16]. Datasets were linked using unique encoded identifiers and analyzed at ICES; the only author who

had access to information which could potentially identify individual participants was DK, an ICES analyst. Data were retrieved on 07/12/2023.

**Outcomes.** GDM screening completion within 20 weeks of the index event (childbirth) satisfied our outcome definition. GDM screening was assessed as a binary variable (completed/not completed) based on individual or combined testing modalities. Individuals may be represented in several outcome groups if they underwent different testing modalities; if an individual received the same testing modality more than once, it was recorded as 'completed' for that modality. Persons who gave birth in 2019 could have completed screening aligned with the alternate pathways introduced during COVID-19 due to retrospective categorization. We assessed screening completion using physician remuneration claims (fee codes) within the Ontario Laboratories Information System (OLIS) and Ontario Health Insurance Plan (OHIP) datasets (S3 File).

**Analysis.** Sociodemographic characteristics of participants were examined along with perinatal health services information. We reported measures of general frequency, central tendency, and dispersion. Data were examined for completeness, outliers, and variable distribution (e.g., skew). Missing demographic data (<1.5% for rurality and CIMD measures) were replaced by simple imputation using predictive mean matching. Chi-square tests enabled comparisons of screening rates across the birth groups. We used unadjusted logistic regression to estimate associations between screening completion and birth year–overall and by risk category. Relative risks were computed using log binomial models. Statistical significance was determined by an alpha level of 0.05. Quantitative analyses were conducted using Statistical Analysis Software, version 9.4.

## Qualitative strand

Qualitative description was used to understand how people who gave birth during the early years of the COVID-19 pandemic experienced GDM screening in Ontario.

**Participants and recruitment.** Eligible participants were those aged 18 or older who gave birth to a live baby in Ontario from May 1, 2020 to December 31, 2021, and were able to participate in an interview conducted in English. Participants were recruited via social media posts to pregnancy and parenting groups. We also placed physical advertisements in public places frequented by new parents (e.g., libraries, pharmacies, recreation centres, early childhood centers). A $25 honorarium was offered. Interviews were conducted July 2022-August 2023.

Participants were sampled for maximum variation in a diverse range of factors, including age, parity, race, geographical location, and socio-economic status. Purposive sampling was used to fill gaps in initial sampling and to expand or add theoretically relevant experiences.

**Data collection.** Semi-structured interviews were conducted via telephone or Zoom based on participants' preferences. Interviews lasted between 19 and 59 minutes (average 41:35) and were conducted by female-identified interviewers who were unknown to participants. All interviews were audio-recorded and transcribed verbatim. The interview guide (S4 File) was collaboratively developed with clinicians, researchers, and those who had lived experiences of pandemic pregnancies. Interview guides were piloted and refined iteratively, and asked participants if they were offered GDM screening and about their overall experience, if they chose to participate.

Data sufficiency was determined by using the model of information power [20], which offers a process for considering adequacy based on the heterogeneity of the sample, focus of the research question, quality of dialogue, and analytic process.

**Analysis.** Our analysis used a qualitative descriptive approach where codes were generated inductively through discussions among the research team. Analytic rigor was ensured by

conducting formal comparisons of themes across participants and interpreting findings through discussions among the team which included clinicians, researchers, and people with lived experience [21]. Qualitative data were managed and analyzed using NVivo version 12.

### Data integration

Data integration occurred in both design and interpretation phases. The project was designed using an objective framework-based mixed methods approach [22]. Following independent data collection and analysis, both quantitative and qualitative results were merged in a visual joint display [23].

**Ethics.** Ethical approval for the use of quantitative health administrative data was waived by the Hamilton Integrated Research Ethics Board (HiREB) based on the use of ICES data in compliance with section 45 of Ontario's Personal Health Information Protection Act (PHIPA). Ethical approval for the qualitative data was granted by HiREB (14632). All qualitative participants provided verbal and written informed consent.

## Results

In the quantitative strand, we identified 85,228 eligible individuals who gave birth from January 1 to March 31 in 2019 (n = 28,642), 2021 (n = 28,753), and 2022 (n = 27,833) (Table 1; Fig 3). The mean age of the included sample was 31 years, prenatal care was primarily provided by obstetricians (65.0%) and midwives (16.8%), and individuals resided in areas characterized by high levels of ethnocultural diversity (28.3%) and residential instability (23.1%).

In the qualitative strand, we collected data from 43 people who had a live birth in Ontario between May 2020 and December 2021 (Table 2). The mean age of interview participants was 31 years, 25 (58.1%) were primiparous, and obstetricians primarily provided prenatal care (88.3%).

### GDM screening participation

Across all GDM screening modalities, the majority of individuals in the quantitative dataset (90.1%) were screened for GDM during pregnancy (Table 3); this significantly increased from 87.4% in 2019 to 91.7% in 2022 (p < .0001). Qualitative data indicated that participation was due in part to belief in the importance of GDM screening, with some describing it as a requirement or obligation.

> *"I think it was a mandatory thing you have to do because they want to make sure, you know, your baby's healthy too, right? If you don't watch it, it could cause too much sugar going to the baby, which is not good"*

(P32, 2021 birth group).

In these descriptions, participants cared for by physicians consistently emphasized that they viewed GDM screening as a directive: *"I wasn't really given the option whether or not to do it. I was just kind of told to do it that I had to get it done"* (P14, 2021 birth group).

Participants receiving care from midwives more often described being offered a choice to participate in GDM screening. Both sets of participants expressed comfort with the style of offer they received and described a commitment to testing based on an understanding of the purpose and significance of the test.

**Table 1. Sociodemographic and perinatal health services characteristics of pregnant persons by birth group (N = 85,228).**

| | N (%) [a] | | | | |
|---|---|---|---|---|---|
| | Overall (N = 85,228) | 2019 Birth Group (n = 28,642) | 2021 Birth Group (n = 28,753) | 2022 Birth Group (n = 27,833) | P value |
| **Sociodemographic characteristics** | | | | | |
| **Age, years** | 31.37 (5.0) | 31.19 (5.1) | 31.28 (4.9) | 31.66 (5.0) | <.0001* |
| **Parity** | | | | | |
| 0 | 37,318 (43.8) | 12,224 (42.7) | 12,804 (44.5) | 12,290 (44.2) | 0.0008* |
| 1 | 30,802 (36.1) | 10,552 (36.8) | 10,231 (35.6) | 10,019 (36.0) | |
| 2 | 10,976 (12.9) | 3,759 (13.1) | 3,655 (12.7) | 3,562 (12.8) | |
| 3 or more | 6,132 (7.2) | 2,107 (7.4) | 2,063 (7.2) | 1,962 (7.0) | |
| **Rural residence (yes)** | 6,111 (7.2) | 2,060 (7.2) | 2,041 (7.1) | 2,010 (7.2) | 0.8376 |
| **Ethnocultural composition [b]** | | | | | |
| 1, least diversity | 13,349 (15.7) | 4,500 (15.7) | 4,544 (15.8) | 4,305 (15.5) | 0.018* |
| 2 | 13,775 (16.2) | 4,531 (15.8) | 4,718 (16.4) | 4,526 (16.3) | |
| 3 | 15,276 (17.9) | 5,039 (17.6) | 5,138 (17.9) | 5,099 (18.3) | |
| 4 | 18,706 (21.9) | 6,272 (21.9) | 6,246 (21.7) | 6,188 (22.2) | |
| 5, greatest diversity | 24,122 (28.3) | 8,300 (29.0) | 8,107 (28.2) | 7,715 (27.7) | |
| **Residential instability [b]** | | | | | |
| 1, least instability | 12,413 (14.6) | 4,130 (14.4) | 4,293 (14.9) | 3,990 (14.3) | 0.0005* |
| 2 | 16,119 (18.9) | 5,292 (18.5) | 5,499 (19.1) | 5,328 (19.1) | |
| 3 | 18,709 (22.0) | 6,317 (22.1) | 6,215 (21.6) | 6,177 (22.2) | |
| 4 | 18,278 (21.4) | 6,044 (21.1) | 6,158 (21.4) | 6,076 (21.8) | |
| 5, greatest instability | 19,709 (23.1) | 6,859 (23.9) | 6,588 (22.9) | 6,262 (22.5) | |
| **Economic dependency [b]** | | | | | |
| 1, least dependency | 24,518 (28.8) | 8,205 (28.6) | 8,283 (28.8) | 8,030 (28.9) | 0.0474 |
| 2 | 17,792 (20.9) | 5,874 (20.5) | 6,054 (21.1) | 5,864 (21.1) | |
| 3 | 15,878 (18.6) | 5,425 (18.9) | 5,190 (18.1) | 5,263 (18.9) | |
| 4 | 14,546 (17.1) | 4,888 (17.1) | 4,965 (17.3) | 4,693 (16.9) | |
| 5, greatest dependency | 12,494 (14.7) | 4,250 (14.8) | 4,261 (14.8) | 3,983 (14.3) | |
| **Infants aged 24 months or less in household (yes)** | 11,231 (13.2) | 3,811 (13.3) | 3,765 (13.1) | 3,655 (13.1) | 0.7277 |
| **Perinatal health services characteristics** | | | | | |
| **Count of early prenatal visits [c]** | 5.11 (3.1) | 5.70 (3.2) | 4.77 (2.9) | 4.87 (3.0) | <.0001* |
| **Prenatal care provider** | | | | | |
| Family Physician | 12,181 (14.3) | 4,404 (15.4) | 3,937 (13.7) | 3,840 (13.8) | <.0001* |
| Midwife | 14,293 (16.8) | 4,737 (16.5) | 4,766 (16.6) | 4,790 (17.2) | |
| Obstetrician | 55,402 (65.0) | 18,484 (64.5) | 18,903 (65.7) | 18,015 (64.7) | |
| Shared care [d] | 2,242 (2.6) | 730 (2.5) | 746 (2.6) | 766 (2.8) | |
| No care/missing/other provider | 1,110 (1.3) | 287 (1.0) | 401 (1.4) | 422 (1.5) | |

[a] Presented as the mean and standard deviation for continuous variables.

[b] Measured as quintiles.

[c] Early visits were defined as occurring prior to 32 weeks of gestation.

[d] Includes patients with an equal number of minor prenatal assessment billings by family physicians and obstetricians between the conception and index date.

*"So, I did take the test. I wanted it so bad because you never know. It's dangerous if you have it and if it's undiagnosed, so yeah. I wanted to get it done and thank god I was negative, I didn't have it."*

(P34, 2021 birth)

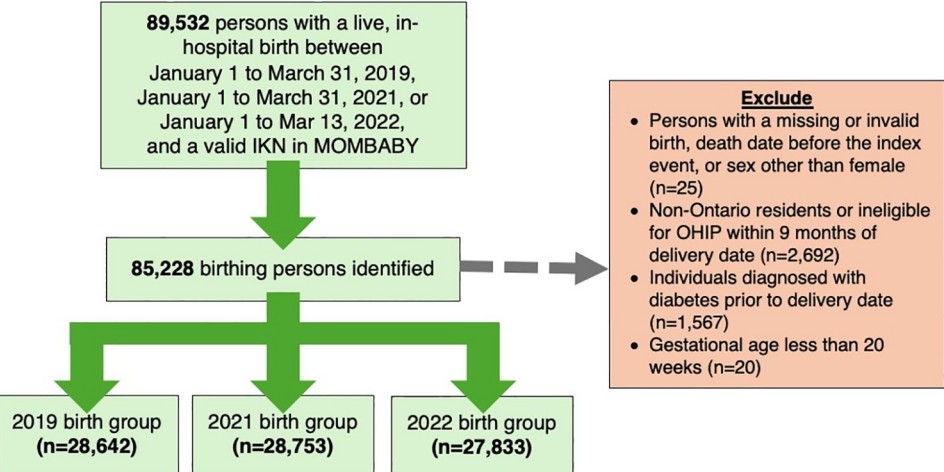

**Fig 3. Flow diagram of quantitative study participants.** IKN = ICES Key Number; MOMBABY = ICES-derived cohort of birthing persons and newborns; OHIP = Ontario Health Insurance Plan.

## GDM screening modalities

Use of the alternate screening pathway for persons without increased risk was significantly higher in 2021 (11.0%) and 2022 (9.0%) than in 2019 (3.2%) (p < .0001) (Table 3). For those with an HbA1c and non-fasting random plasma glucose result above the cut-off, the rate of completing an OGCT and OGTT (as per the alternate pathway) remained the same over time; 0.1% of persons in all study years satisfied this screening pathway (p = 0.2639). Despite the adoption/rise of these alternate screening strategies, traditional modalities remained frequent for each birth cohort (OGCT only: 2019, 75.7%; 2021, 71.8%; 2022, 74.1%; p < .0001 | OGCT followed by OGTT: 2019, 6.1%; 2021, 6.0%; 2022, 6.8%; p = 0.0003).

Participants in the qualitative dataset reported different experiences and opinions of adapted screening pathways. Participants who underwent the adapted pathways understood that their healthcare provider suggested these techniques to minimize wait times and reduce their COVID-19 exposure risks, but some questioned the effectiveness of these adapted pathways:

> *"The first time they just did a blood draw. No fasting, blood draw, no drink. Because of COVID, to minimize the amount of time I'd have to wait, not just me but patients in the practice would have to wait in the waiting area, I guess. So, they did that, but then later in my pregnancy, they were concerned I might have it. So, at like, 3 weeks before I delivered, I did the two-hour glucose test at that point. It was borderline positive. So, who knows if I had it earlier and it was missed—or if I developed it later. Who knows"*

(P15, 2021 birth group).

Although participants felt that they had sufficient knowledge to provide informed consent, most reported that they were not given a choice regarding which testing pathway they would participate in.

> *"They did not give me the option to do the alternative* [HbA1c instead of OGTT] *at the beginning. I don't think I asked, I mean, I was happy to not have to do it, because I'd heard*

**Table 2. Baseline characteristics of interview participants (n = 43).**

| | N (%) [a] |
|---|---|
| **Sociodemographic characteristics** | |
| **Age at birth, years** | 31.4 (4.3) |
| **Parity** | |
| Multiparous | 15 (34.9) |
| Primiparous | 25 (58.1) |
| **Ethnicity** [b] | |
| Black | 2 (4.7) |
| East Asian (Asian) | 1 (2.3) |
| East Asian (Chinese) | 3 (7.0) |
| South Asian (Bangladeshi) | 1 (2.3) |
| White | 36 (83.7) |
| **Residence** | |
| Rural | 7 (16.3) |
| Suburban | 14 (32.6) |
| Urban | 22 (51.2) |
| **Number of people living in the household at time of birth** | |
| 2 | 17 (39.5) |
| 3 | 13 (30.2) |
| 4 | 6 (14.0) |
| 5 or more | 7 (16.3) |
| **Education status** | |
| Completed postgraduate or professional studies | 11 (25.6) |
| Some postgraduate or professional studies | 3 (7.0) |
| Completed university | 17 (39.5) |
| Completed college or trade school | 8 (18.6) |
| Some college or trade school | 3 (7.0) |
| Completed high school | 1 (2.3) |
| **Prenatal health service characteristics** | |
| **Birth year** | |
| 2020 | 19 (44.2) |
| 2021 | 24 (55.8) |
| **Planned pandemic pregnancy** | |
| No | 19 (44.2) |
| Yes | 24 (55.8) |
| **Prenatal care provider** | |
| Family physician | 16 (37.2) |
| Midwife | 10 (23.2) |
| Obstetrician | 38 (88.3) |

[a] Presented as the mean and standard deviation for continuous variables.

[b] Categorized based on guidance from the Canadian Institute of Health Information.

*it wasn't pleasant, and now* [with a later pregnancy] *I can confirm it is. But, yeah, they just, maybe if I'd asked, they would've given me the choice. I'm not sure, but it wasn't presented*"

(P15, 2021 birth group).

**Table 3. Gestational diabetes screening completion among pregnant persons in the primary study cohort, by birth group.**

| | N (%) | | | | |
|---|---|---|---|---|---|
| | Overall n = 85,228 | 2019 Birth Group [a] n = 28,642 | 2021 Birth Group n = 28,753 | 2022 Birth Group n = 27,833 | P value [b] |
| **Overall screening** | | | | | |
| **GDM screening completed using any testing modality** | 76,814 (90.1) | 25,037 (87.4) | 26,253 (91.3) | 25,524 (91.7) | <.0001* |
| **Screening modalities by pathway** | | | | | |
| **Usual screening pathway during minimal health system disruptions** | | | | | |
| OGCT only | 62,963 (73.9) | 21,692 (75.7) | 20,644 (71.8) | 20,627 (74.1) | <.0001* |
| OGCT followed by OGTT | 5,363 (6.3) | 1,752 (6.1) | 1,727 (6.0) | 1,884 (6.8) | 0.0003* |
| **Temporary alternate screening pathway during moderate or severe health system disruptions [c]** | | | | | |
| HbA1c and non-fasting random plasma glucose test | 6,604 (7.7) | 926 (3.2) | 3,171 (11.0) | 2,507 (9.0) | <.0001* |
| HbA1c and non-fasting random plasma glucose test followed by OGCT and OGTT | 64 (0.1) | 16 (0.1) | 22 (0.1) | 26 (0.1) | <.0001* |

GDM = gestational diabetes mellitus; OGCT = oral glucose challenge test; OGTT = oral glucose tolerance test; HbA1c = Hemoglobin A1C.

* Significant at the level of p<0.05.

[a] The inclusion of participants who gave birth in 2019 in the COVID-19 alternate screening pathway is due to retrospective categorization, where previously utilized alternative screening methods have been reclassified under this strategy for consistent data analysis.

[b] Chi-square test comparing screening completion across the three birth groups.

[c] The modalities below apply to persons without increased risk of GDM. For screening among high-risk persons during times of moderate or severe health system disruptions, refer to OGCT only and OGCT followed by OGTT.

Some midwifery clients were presented with a choice of testing modality and typically demonstrated clear knowledge of the options and described themselves as making an informed decision about whether to participate in GDM screening and which modality to choose:

*"Mostly because I found getting a blood draw less invasive than how I would potentially feel drinking that sugary drink like I felt like it would make me sick. So, I did the blood–I was very aware though if I had to do it I would do the drink. But if I could challenge it with just the blood draw, I thought I would do that because it was–I was able to go I think that day I went for my midwife appointment to the blood clinic, it's all in one building. So, I just went did it that day and was good to go then."*

(2021 birth group, p#65)

## Factors associated with alternate GDM screening modalities during COVID-19

Completion of HbA1c and non-fasting random plasma glucose test for patients without increased risk was significantly greater in 2021 (RR = 3.41; 95% CI = 3.18–3.66) and 2022 (RR = 2.79; 95% CI = 2.59–3.00) compared to 2019 (Table 4). For high-risk persons, OGCT followed by OGTT completion was significantly greater in 2022 compared to 2019 (RR = 1.11; 95% CI = 1.04–1.18). The risk of completing only an OGCT among high-risk persons decreased compared to 2019 (2021: RR = 0.95, 95% CI = 0.94–0.96 | 2022: RR = 0.98, 95% CI = 0.97–0.99).

**Table 4. Factors associated with alternate GDM screening completion during COVID-19.**

| | RR (95% CI) | | | |
|---|---|---|---|---|
| Clinical indication | No increased risk of GDM | | High-risk persons [a] | |
| GDM screening modality | HbA1c and non-fasting random plasma glucose test | HbA1c and non-fasting random plasma glucose test followed by OGCT and OGTT | OGCT only | OGCT followed by OGTT |
| **Birth year** | | | | |
| 2019 (ref) | - | - | - | - |
| 2021 | 3.41 (3.18–3.66) | 1.37 (0.72–2.61) | 0.95 (0.94–0.96) | 0.98 (0.92–1.05) |
| 2022 | 2.79 (2.59–3.00) | 1.67 (0.90–3.12) | 0.98 (0.97–0.99) | 1.11 (1.04–1.18) |

RR = relative risk; CI = confidence interval; GDM = gestational diabetes mellitus; OGCT = oral glucose challenge test; OGTT = oral glucose tolerance test; HbA1c = Hemoglobin A1C.

[a] Based on the presence of at least one: Previous GDM, high-risk ethnicity, aged ≥40, or obesity.

## GDM screening experiences

Participation in GDM screening was described as stressful by some participants who under-stood that attendance at a medical laboratory may expose them to COVID-19. However, they consistently described the importance of screening and expressed a commitment to participating, even when participation meant increasing their risk of COVID-19 exposure:

*"I think I just wanted to go for it because I know how dangerous gestational diabetes can be if it's not monitored. So, for me, the benefits outweigh the risks because gestational diabetes is very dangerous if it's not caught"*

(P07, 2021 birth group).

Participants also discussed efforts that they and others made to minimize the potential of exposure to COVID-19 and balance the integrity of the GDM screening test. Some participants recounted being asked to wait for the next screening step outside the testing facility (i.e., in a parking lot), but others were not offered these measures, and a few were refused when they made similar requests:

*"It was very stressful because I had to go take it in* [nearby city]*, which, at the time, had the most COVID cases out of everywhere. And when I asked the lab if I could go wait the hour outside in my car which was right outside the window of the place, they told me no. And then they made me sit in the lab in the waiting room for an hour, and I was like very angry and very stressed out after that"*

(P30, 2020 birth group).

Access to GDM was a challenge for some participants in times when the healthcare system was heavily burdened with COVID-19. Due to reduced laboratory capacity, as COVID-19-related services were prioritized, some participants found it difficult to book a screening appointment during the optimal gestational window. For example, P1 shared:

*"It was challenging to book that* [GDM] *test because none of the labs—like, for example,* [a medical laboratory chain] *where you can have these tests done, they had very limited*

*appointments. So, you couldn't get an appointment for 3 or 4 weeks, but you needed to have that glucose test at a certain week of your pregnancy"*

(2021 birth group).

## Discussion

This mixed methods study examined changes in GDM screening rates and modalities among people who gave birth in Ontario hospitals in 2019, 2021, and 2022. Administrative data were complemented by qualitative interviews about pregnant persons' experiences with GDM screening. Importantly, this study demonstrated that despite major health system challenges posed by the COVID-19 pandemic, GDM screening rates significantly increased. Our findings of increased screening rates align with trends observed elsewhere in Canada, including over 95% of pregnant individuals being screened for GDM during the pandemic in Alberta [24]. They conflict with a study conducted in the United States, which reported a slight decrease in GDM screening rates during COVID-19 compared to pre-pandemic [13].

Our study shows that this high rate of GDM screening participation may have been achieved by changing the modality of testing, permitting the continued health system prioritization of GDM identification and treatment [12]. The proportion of pregnant persons in Ontario who completed the alternate screening pathway (an HbA1c and non-fasting random plasma glucose test) increased by about 6–8% in 2021 and 2022 compared to 2019. Although there were some variations in rates for the usual screening pathways (OGCT only or OGCT followed by OGTT), this may be attributed to the evolving screening guidance introduced for high-risk persons in late 2021 [12]. These adaptations may have helped ensure continued GDM screening practices at a time when the pandemic severely strained laboratory testing facilities and clinician capacities [12]. That guidance recognized that the initial GDM adaptation was proposed at a time when the impact of the COVID-19 pandemic on prenatal care access and laboratory services was largely unknown. With a clearer understanding of these challenges, a different tactic for GDM screening could be taken to try and provide flexibility depending on the pandemic burden while still prioritizing GDM identification. The swift issuance of guidance and later updates also provided timely guidance to clinicians about prioritizing GDM screening, even amongst other health system priorities. This prioritization by clinicians may have contributed to counselling, which supported patient perception of GDM screening as a crucial part of prenatal care. Pregnant individuals in this study and others widely recognized and emphasized the importance of GDM screening, owing to the severity of the disease and the likelihood that a patient may not notice initial onset of diabetes without a screening test [25].

Our findings highlight increased adoption of alternative screening modalities during a time of public health emergency, reflecting robust compliance to adapted GDM guidelines that were implemented in response to the pandemic [26]. The increase in adoption may reflect the success of the adapted screening pathways in maintaining the accessibility of GDM screening to patients throughout the pandemic, in part by allowing latitude to adapt the modality offered to patient risk and local pandemic burden. Maintaining high levels of GDM screening uptake facilitated the development of other management adaptations for those diagnosed with GDM (27). While our study demonstrates that these adapted guidelines ensured continued participation in GDM screening in a resource-strained environment, this analysis cannot yield information on the efficacy of these adaptations, an issue raised by a small number of participants in this and other qualitative studies [27]. Clinicians and scientists have widely debated the

efficacy of adapted screening methods, and evidence on this topic is emerging [27–31]. High screening uptake was, in part, explained by the qualitative finding that participants consistently understood GDM screening to be highly important or even obligatory. Most participants perceived themselves as consenting to their clinicians' choices (i.e., whether to screen, how to screen) rather than actively making or sharing in these decisions. While participants in the current study widely endorsed an offer of GDM screening that conformed to the principles of informed consent, participants in some other jurisdictions have indicated a preference to choose their own GDM screening modality [27,32]. The variety of preferences for decision-making about GDM screening emphasizes the complex nature of these decisions. Decisions about GDM screening modality rely on specialized clinical knowledge about personal risk, laboratory capacity, and test sensitivity and specificity. Patient values about tolerance of risk associated with COVID-19 exposure, and willingness to tolerate false positive or negative results when choosing a GDM screening modality provide important input.

## Implications

While alternative screening strategies for GDM have proven beneficial amidst challenges posed by the COVID-19 pandemic, their comparative effectiveness against traditional screening methods warrants additional scrutiny [30–33]. The efficacy, sensitivity, and overall patient outcomes associated with these alternative methods require thorough investigation to ensure optimal care standards are maintained. Furthermore, patient education and active involvement in healthcare decisions are important, especially given the intricate and rapidly shifting dynamics observed in healthcare settings amidst crises such as the COVID-19 pandemic.

## Limitations

Our quantitative cohort was restricted to individuals who gave birth during specific three-month birthing windows to lessen the effects of confounding by eliminating overlapping gestational periods; this may limit the generalizability of findings. The chosen birth periods may not fully capture how changes to and trends in GDM screening practices evolved during the COVID-19 pandemic. Our restriction to in-hospital births and live births may have excluded those at the highest risk of non-screening for GDM. Sociodemographic factors were only available via neighbourhood-level information within our administrative data source. Qualitative data gathered between July 2022 and August 2023 (from individuals who gave birth between May 2020 and December 2021) may have been subject to recall bias about whether and how a choice of testing modality was offered.

## Conclusion

Our mixed-methods study of GDM screening uptake and experiences during the COVID-19 pandemic revealed that even in the face of health system disruptions, screening participation increased in live hospital-birthing people, highlighting the adaptability of the healthcare system and commitment to maintaining essential prenatal care. The adapted screening guidance successfully ensured wide access to GDM screening, and patients were generally pleased to accept their clinicians' recommendations, endorsing a model of informed consent for this form of screening. The adapted GDM guidelines offer an example of how the healthcare system effectively pivoted in a time of crisis to meet the needs of pregnant patients within a resource-constrained environment. Our study contributes to the growing body of knowledge on prenatal care and GDM screening and offers a foundation for future research and policy development to optimize care for pregnant individuals.

## Supporting information

**S1 File. Methodological reporting checklists.**
(DOCX)

**S2 File. Timeline of lookback windows for outcomes of interest, by birth group.**
(DOCX)

**S3 File. Outcomes and corresponding datasets.**
(DOCX)

**S4 File. Qualitative interview guide.**
(DOCX)

## Acknowledgments

This document used data adapted from the Statistics Canada Postal Code[OM] Conversion File, which is based on data licensed from Canada Post Corporation, and/or data adapted from the Ontario Ministry of Health Postal Code Conversion File, which contains data copied under license from ©Canada Post Corporation and Statistics Canada. Parts of this material are based on data and/or information compiled and provided by the Canadian Institute for Health Information and the Ontario Ministry of Health. The analyses, conclusions, opinions and statements expressed herein are solely those of the authors and do not reflect those of the funding or data sources; no endorsement is intended or should be inferred. We acknowledge Caroline Mniszak's contributions to the interview guide and recruitment strategies.

## Author Contributions

**Conceptualization:** Rebecca H. Correia, Sarah D. McDonald, Elizabeth K. Darling, Aaron Jones, Michelle Howard, Devon Greyson, Meredith Vanstone.

**Data curation:** Dima Hadid, Rebecca H. Correia, David Kirkwood, Andrea Carruthers, Cassandra Kuyvenhoven.

**Formal analysis:** Dima Hadid, Rebecca H. Correia, David Kirkwood, Andrea Carruthers, Cassandra Kuyvenhoven.

**Funding acquisition:** Sarah D. McDonald, Elizabeth K. Darling, Michelle Howard, Devon Greyson, Sujane Kandasamy, Meredith Vanstone.

**Methodology:** Dima Hadid, Rebecca H. Correia, Sarah D. McDonald, Elizabeth K. Darling, David Kirkwood, Aaron Jones, Meredith Vanstone.

**Supervision:** Devon Greyson, Meredith Vanstone.

**Validation:** Sarah D. McDonald, Elizabeth K. Darling, Aaron Jones, Michelle Howard, Devon Greyson, Sujane Kandasamy, Meredith Vanstone.

**Writing – original draft:** Dima Hadid, Rebecca H. Correia.

**Writing – review & editing:** Sarah D. McDonald, Elizabeth K. Darling, David Kirkwood, Aaron Jones, Andrea Carruthers, Cassandra Kuyvenhoven, Michelle Howard, Devon Greyson, Sujane Kandasamy, Meredith Vanstone.

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
