## [Decision Letter · Decision Letter 0]

11 Nov 2024

PONE-D-24-41745Assessing the impact of the COVID-19 pandemic on uptake and experiences of Gestational Diabetes Mellitus Screening in Ontario: A parallel convergent mixed-methods studyPLOS ONE

Dear Dr. Vanstone,

Thank you for submitting your manuscript to PLOS ONE. After careful consideration, we feel that it has merit but does not fully meet PLOS ONE’s publication criteria as it currently stands. Therefore, we invite you to submit a revised version of the manuscript that addresses the points raised during the review process.

Please include more discussions of the results to make the discussion section more robust as reviewers suggested.

We look forward to receiving your revised manuscript.

Kind regards,

Dongmei Li

Academic Editor

PLOS ONE

Journal Requirements: When submitting your revision, we need you to address these additional requirements. 1. Please ensure that your manuscript meets PLOS ONE's style requirements, including those for file naming. The PLOS ONE style templates can be found at https://journals.plos.org/plosone/s/file?id=wjVg/PLOSOne_formatting_sample_main_body.pdf and https://journals.plos.org/plosone/s/file?id=ba62/PLOSOne_formatting_sample_title_authors_affiliations.pdf 2. We note that the grant information you provided in the ‘Funding Information’ and ‘Financial Disclosure’ sections do not match.  When you resubmit, please ensure that you provide the correct grant numbers for the awards you received for your study in the ‘Funding Information’ section. 3. Thank you for stating the following financial disclosure: "This study was funded by the Canadian Institutes of Health Research grant number 179921. MV is supported by a Canada Research Chair in Ethical Complexity in Primary Care. SM is supported by a Canada Research Chair in Maternal and Child Obesity Prevention and Intervention. DG is supported by a CIHR/PHAC Applied Public Health Chair and Michael Smith Health Research BC Scholar Award." Please state what role the funders took in the study.  If the funders had no role, please state: ""The funders had no role in study design, data collection and analysis, decision to publish, or preparation of the manuscript."" If this statement is not correct you must amend it as needed. Please include this amended Role of Funder statement in your cover letter; we will change the online submission form on your behalf. 4. Thank you for stating the following in the Acknowledgments Section of your manuscript: "This study was supported by ICES, which is funded by an annual grant from the Ontario  Ministry of Health (MOH) and the Ministry of Long-Term Care (MLTC). This document used data adapted from the Statistics Canada Postal CodeOM 416 Conversion File, which is based on data  licensed from Canada Post Corporation, and/or data adapted from the Ontario Ministry of Health  Postal Code Conversion File, which contains data copied under license from ©Canada Post  Corporation and Statistics Canada. Parts of this material are based on data and/or information  compiled and provided by the Canadian Institute for Health Information and the Ontario  Ministry of Health. The analyses, conclusions, opinions and statements expressed herein are  solely those of the authors and do not reflect those of the funding or data sources; no  endorsement is intended or should be inferred. We acknowledge Caroline Mniszak’s  contributions to the interview guide and recruitment strategies." We note that you have provided funding information that is not currently declared in your Funding Statement. However, funding information should not appear in the Acknowledgments section or other areas of your manuscript. We will only publish funding information present in the Funding Statement section of the online submission form. Please remove any funding-related text from the manuscript and let us know how you would like to update your Funding Statement. Currently, your Funding Statement reads as follows: "This study was funded by the Canadian Institutes of Health Research grant number 179921. MV is supported by a Canada Research Chair in Ethical Complexity in Primary Care. SM is supported by a Canada Research Chair in Maternal and Child Obesity Prevention and Intervention. DG is supported by a CIHR/PHAC Applied Public Health Chair and Michael Smith Health Research BC Scholar Award." Please include your amended statements within your cover letter; we will change the online submission form on your behalf. 5. We noted in your submission details that a portion of your manuscript may have been presented or published elsewhere. "None of the article contents are under consideration for publication elsewhere or have been published in any journal. We have published other parts of the interview data (pertaining to different topics - feelings of depression and anxiety and experiences of post-partum hospital stays) in the journals Women’s Health and Journal of Obstetrics and Gynecology Canada. No portion of the text has been copied from other material in the literature unless identified with quotation marks or a citation." Please clarify whether this conference proceeding or publication was peer-reviewed and formally published. If this work was previously peer-reviewed and published, in the cover letter please provide the reason that this work does not constitute dual publication and should be included in the current manuscript. 6. Please review your reference list to ensure that it is complete and correct. If you have cited papers that have been retracted, please include the rationale for doing so in the manuscript text, or remove these references and replace them with relevant current references. Any changes to the reference list should be mentioned in the rebuttal letter that accompanies your revised manuscript. If you need to cite a retracted article, indicate the article’s retracted status in the References list and also include a citation and full reference for the retraction notice.

Reviewers' comments:

Reviewer's Responses to Questions

**Comments to the Author**

1. Is the manuscript technically sound, and do the data support the conclusions?

Reviewer #1: Yes

Reviewer #2: Yes

2. Has the statistical analysis been performed appropriately and rigorously? 

Reviewer #1: Yes

Reviewer #2: Yes

3. Have the authors made all data underlying the findings in their manuscript fully available?

Reviewer #1: Yes

Reviewer #2: No

4. Is the manuscript presented in an intelligible fashion and written in standard English?

Reviewer #1: Yes

Reviewer #2: Yes

5. Review Comments to the Author

Reviewer #1: Thanks for good and reliable data of this important topic. The paper is complete in every section with sufficient information. Please explain more about the reason that you chose this topic. In title: is assessing or evaluating?

Reviewer #2: A well planned, carefully conducted convergent mixed-methods study that provides valuable insights into the effects of the COVID-19 pandemic on gestational diabetes screening. Following series of reporting guidelines, the authors examine the changes in uptake, modality, and experiences of GDM screening in Ontario, Canada during the COVID-19 pandemic. Authors conclude that despite health system challenges, GDM screening rates increased in the study population, demonstrating the success of adapted GDM screening guidelines.

Below are some recommendations.

Their paper is well written, however, I feel that the discussion around the differences between each year metrics needs to be expanded in terms of possible explanations for the differences and accounting for the introduction of modifications to the diagnostic algorithm in a chronological order (as mentioned at Line 354 'this may be attributed to the evolving screening guidance introduced for high-risk persons in late 2021') This will give better understanding of why the authors have arrived at their conclusions.

6. PLOS authors have the option to publish the peer review history of their article (what does this mean?). If published, this will include your full peer review and any attached files.

Reviewer #1: No

Reviewer #2: **Yes: **Diego J. Aparcana-Granda

---

## [Author Response · Author response to Decision Letter 0]

28 Nov 2024

We have uploaded a response to specific reviewer and editor comments, see attached file.

---

## [Editor Report · Decision Letter 1]

4 Dec 2024

Assessing the impact of the COVID-19 pandemic on uptake and experiences of Gestational Diabetes Mellitus Screening in Ontario: A parallel convergent mixed-methods study

PONE-D-24-41745R1

Dear Dr. Vanstone,

We’re pleased to inform you that your manuscript has been judged scientifically suitable for publication and will be formally accepted for publication once it meets all outstanding technical requirements.

Kind regards,

Dongmei Li

Academic Editor

PLOS ONE
---

## [Editor Report · Acceptance letter]

16 Dec 2024

PONE-D-24-41745R1 

PLOS ONE

Dear Dr. Vanstone, 

I'm pleased to inform you that your manuscript has been deemed suitable for publication in PLOS ONE. Congratulations! Your manuscript is now being handed over to our production team.

Kind regards, 

on behalf of

Dr. Dongmei Li 

Academic Editor

PLOS ONE